# Systematic Experimental Assessment of POFA Concrete Incorporating Waste Tire Rubber Aggregate

**DOI:** 10.3390/polym14112294

**Published:** 2022-06-05

**Authors:** Akram M. Mhaya, S. Baharom, Mohammad Hajmohammadian Baghban, Moncef L. Nehdi, Iman Faridmehr, Ghasan Fahim Huseien, Hassan Amer Algaifi, Mohammad Ismail

**Affiliations:** 1Department of Civil Engineering, Universiti Kebangsaan Malaysia (UKM), Bangi 43600, Selangor, Malaysia; akrammhaya1985@gmail.com; 2Department of Manufacturing and Civil Engineering, Norwegian University of Science and Technology (NTNU), 2815 Gjøvik, Norway; 3Department of Civil Engineering, McMaster University, Hamilton, ON L8S 4M6, Canada; 4Institute of Architecture and Construction, South Ural State University, Lenin Prospect 76, 454080 Chelyabinsk, Russia; s.k.k-co@live.com; 5Department of the Built Environment, School of Design and Environment, National University of Singapore, Singapore 117566, Singapore; bdggfh@nus.edu.sg; 6Faculty of Civil and Environmental Engineering, Universiti Tun Hussein Onn Malaysia, Parit Raja 86400, Johor, Malaysia; enghas78@gmail.com; 7School of Civil Engineering, Faculty of Engineering, Universiti Teknologi Malaysia, Skudai 81310, Johor, Malaysia

**Keywords:** rubberized concrete, POFA, waste, tire rubber, artificial neural network, sustainability

## Abstract

Several researchers devoted considerable efforts to partially replace natural aggregates in concrete with recycled materials such as recycled tire rubber. However, this often led to a significant reduction in the compressive strength of rubberized concrete due to the weaker interfacial transition zone between the cementitious matrix and rubber particles and the softness of rubber granules. Thereafter, significant research has explored the effects of supplementary cementitious materials such as zeolite, fly ash, silica fume, and slag used as partial replacement for cement on rubberized concrete properties. In this study, systematic experimental work was carried out to assess the mechanical properties of palm oil fuel ash (POFA)-based concrete incorporating tire rubber aggregates (TRAs) using the response surface methodology (RSM). Based on the findings, reasonable compressive, flexure, and tensile strengths were recorded or up to 10% replacement of sand with recycled tire fibre and fine TRAs. In particular, the reduction in compressive, tensile, and flexural strengths of POFA concrete incorporating fibre rubber decreased by 16.3%, 9.8%, and 10.1% at 365 days compared to normal concrete without POFA and rubber. It can be concluded that utilization of a combination of POFA and fine or fibre rubber could act as a beneficial strategy to solve the weakness of current rubberized concrete’s strength as well as to tackle the environmental issues of the enormous stockpiles of waste tires worldwide.

## 1. Introduction

Concrete is a manufactured material that is most widely used since invented; comprises three major elements: binder (supplementary cement materials), aggregates, and water. In suitable proportions, these elements will set and harden over time [1,2,3]. Aggregates being the inert material and major constituent in concrete, play a vital role in accomplishing innovation in the concrete industry [4,5,6]. The fraction of aggregates is about 60% to 80% of the concrete total volume [7,8]. Moreover, the preparation of some types of concrete such as light and heavyweight concrete, as well as concrete resistant to sound or vibration, can be achieved through the proper selection of aggregates [9,10,11,12,13]. In developing almost all types of special concrete, selecting the proper aggregates, and manipulating their size distributions are two important steps. On the other hand, the high consumption of natural sources, together with extreme quantities of industrial wastes being produced, as well as inevitable environmental pollution occurrences, demand the delineation of new solutions for sustainable development [14,15,16].

Waste tire aggregate is non-biodegradable in the short term. The main components waste tire aggregate consists of are the natural and synthetic rubber (also known as a polymer). The styrene-butadiene in the synthetic rubber ratio influences the properties of the polymer: with high styrene content, the rubbers are harder and less rubbery, and takes a longer time to dissolve under normal conditions, which leads to a rise in environmental concern when it ends life service [17,18,19,20,21]. Currently, the worldwide growth of the automobile industry and the increase in car use have tremendously boosted tire production. Annually, around billions of tires have been used and end their service life, and more than 50% are discarded to landfills or garbage without any treatment, continuing to pose environmental challenges. By the year 2030, there would be 5000 million tires to be discarded on a regular basis [22,23]. In this spirit, waste materials and by-product recycling has become an attractive alternative to disposal. [24,25]. For instance, inclusion of tire crumbs has great potential to enhance the acoustic insulation and ductility properties of concrete [26,27,28], and also brings sustainability and environmental benefits [29,30,31]. However, a significant reduction in the strength prevents engineers from fully benefiting from this attribute [32]. This reduction in strength is due to a major weakness of the bond among the cement paste and rubber particles [33,34]. Turatsinze et al. [35] observed that the hydrophobic nature of rubber led to a weak bonding strength of the interfacial transition zone (ITZ) among cement paste and rubber particles. The strength performance of the rubberized concrete was also influenced by the stiffness, particle size, gradation, surface properties, and amount of rubber [36,37]. Meanwhile, some studies recommended using additives or pre-treatment of the rubber to enhance the performance of rubberized concrete [38,39,40,41]. Therefore, a recent development in the field of concrete science is the utilization of waste materials and by-products as supplementary cementitious materials (SCMs) in the construction industry, such as palm oil fuel ash (POFA) [42,43,44]. The use of these materials not only can be recycled as construction materials, but also is able to reduce the cost of cement and concrete manufacturing [45,46,47,48]. As an agricultural by-product, POFA has proved to have great potential to serve as a partial replacement binder material in concrete mixtures [49]. However, by increasing the amount of POFA, the compressive strength decreased due to the higher water demand [50]. Meanwhile, Sata et al. [51] produced 30% ground POFA-modified concrete with high strength by including a superplasticizer (SP) to reduce the water to cement ratio and achieved a higher strength than that of conventional concrete at 28 days. Moreover, the particle size of POFA plays a major role in compressive strength development. POFA with a finer particle size significantly improves the compressive strength of concrete in comparison to medium-ground or unground POFA. The lower strength development of unground POFA is due to the lower surface area, which decreases the pozzolanic reaction of POFA, thus reducing the strength development of concrete. Furthermore, the pore structure of unground POFA is less dense than OPC. Utilization of POFA with high fineness particles tends to form denser concrete which is more homogeneous, thus improving the compressive strength development [52]. This may be attributed to the production of extra C-(A)-S-H gel, which contributed to the densification of the internal structure of concrete by making it stiffer [53,54,55], and it may improve the microstructure, durability, and mechanical properties of mortar and concrete, which may be difficult to achieve using only ordinary Portland cement [56,57].

In addition, the majority of the existing literature relies on the traditional lab work, while the systematic experimental work of rubberized concrete is still in its infant. For example, response surface methodology (RSM) was widely used to not only reduce the number of experimental designs, but also to assess the interaction and relationship between the affecting parameters, as well as to predict and optimize the output effectively [58,59,60]. For instance, Algaifi, Mustafa Mohamed [61] optimize the mechanical properties of alkali-activated mortar incorporating (granulated blast-furnace slag) GBFS, fly ash, and Nano-Silica using RSM. Based on the outcome, it was inferred that the quadratic model predicted quite accurate and reliable results with the highest correlation (*R*^2^) of 0.9. Similarly, the efficiency of regressions and machine-learning (ML) models such as support-vector machine (SVM), genetic programming (GP), and Artificial neural network (ANN), were also developed to evaluate the strength of rubberized concrete [62]. From this perspective, the current study aims at improving the low strength of rubberized concrete by utilizing POFA as an OPC replacement to enhance the bond between the rubber aggregates’ surface and cement paste, as well as to prepare supportive information on the mechanical properties by using a systematic experimental design.

## 2. Materials and Process Technology Standardization

### 2.1. Materials’ Properties

In this study, Ordinary Portland Cement (OPC), Type I, which meets standard EN 197-1-CEM I—42.5 N, was used for this research. The POFA used in this study was obtained from Kahang mill, Kluang Johor, in the south of Malaysia. The POFA, with a particle size less than 200 µm, was collected and transferred to the grounding machine. Figure 1 schematically illustrates the complete procedure of preparing POFA.

The pozzolanic activity of POFA improves with size reduction [63]. Therefore, to obtain the best results, the POFA was ground for 6 hours. Based on ASTM C618-05, the maximum amount of retained materials on a 45 µm No. 325 sieve should not exceed 35%. In general, the main components of POFA and OPC were similar. However, the major compounds of OPC were calcium oxide (CaO) at 68.3%, silicon oxide (SiO_2_) at 16.4%, and aluminum oxide (Al_2_O_3_) at 4.24%. Meanwhile, POFA is rich in SiO_2_ at 63.7%, which is a percentage similar to that of other pozzolanic materials. The results on the characteristics of POFA and OPC are presented in two parts, namely physical properties, and chemical characterization in Table 1.

Natural river sand quartz (fine aggregate) and gravel (coarse aggregate) were used in compliance with ASTM C33 [64] requirements. Sieve analysis was conducted to determine the grading of fine and coarse aggregate according to ASTM C136. Meanwhile, tire rubber aggregate (TRA) is an engineering product made by cutting scrap tires into small pieces using shredders and shearing equipment. The typical physical and chemical specifications of tire particles are presented in Table 2, according to the company statement.

Tire rubber aggregates were categorized into three groups, namely fine fibre crumb rubber (0.8 to 3.3 mm), fine granular crumb rubber (1 to 4 mm), and coarse granular crumb rubber (5 to 8 mm). Fine fibre tire rubber aggregate—Type R_1_ has a maximum length of 16 mm. Fibre rubber aggregate (R_1_) was replaced partially as the natural fine aggregate. Fine granular tire rubber aggregate—Type R_2_ sized was replaced partially as a fine aggregate. Coarse granular tire rubber aggregate—Type R_3_ particle size was replaced as a conventional coarse aggregate (Figure 2). Figure 3 shows the result of sieve analysis of natural and tire rubber aggregate was in range basis on ASTM C33/C 33M that permissible limit is indicated as 1%.

A high-range water reducer was used to achieve the desired workability. The basic component of super plasticiser (SP) is synthetic polymers base, and it is a chloride-free product. The used SP meets the requirement of ASTM C494 for types A and F. This admixture is compatible with all types of cement. This type of SP is normally dispensed at a rate of 0.8 to 3 litres per 100 kilograms of cementitious material. Table 3 shows the typical properties of utilized SP.

### 2.2. Concrete Mix Design for Rubberized Concrete

A 20% level of POFA replacement with 0.38 water–binder ratio and with the introduction of 1% super plasticiser was determined as the optimum SP admixture to satisfy the desired workability from the previous steps of modification mix design. The next phase of modification of the concrete mix design is the introduction of a rubber aggregate to the concrete mixture. There is no standard method or globally accepted method to manage the amount and size of rubber aggregates for rubberized concrete. According to previous experiences, many opinions were given by researchers relative to experimental observations in different studies. Two methods can be used for substituting materials in concrete mixtures: equivalent weight of rubber materials or equivalent absolute volume of rubber materials in the mixture. Both methods have been used in previous studies. Some researchers applied substitute materials based on the weight percentages of the total mixture [31,65,66], whereas some used the equivalent absolute volume of materials [67,68,69]. However, it is worth mentioning that the equivalent absolute volume of rubber materials in the mixture showed results closer to those of conventional concrete. Therefore, for all substitutions of natural aggregates by rubber aggregates, the replacement was conducted by volume and not by mass. The corresponding weights of the calculated volumes were then used. In the American Concrete Institute ACI 211.1 method, mix design concrete was used for designing the mix proportioning of concrete. The target grade of concrete design was chosen as 38. To establish the maximum amount of TRA that can be replaced as fine or coarse aggregate, such that the produced concrete could be considered as a structural application, several trials of mix design were conducted. The mix proportion design was established based on the weight method, as described in ACI 211.1. Table 4 presents the mix proportion of rubberized concrete with varying percentages of rubber with different types and sizes of rubber aggregates.

### 2.3. Experimental Program

Various tests were conducted to identify the normal and POFA-modified rubberized concrete properties. The workability of freshly mixed POFA-rubberized concrete was evaluated through a slump measurement, as outlined in ASTM C143; a slump test was conducted using a frustum of cone for the mould with the following values: 305 mm high; 102 mm diameter at the top; 203 mm diameter at the base. After filling the container with concrete in three layers, each layer was compacted 25 times using a steel rod with a standard 16 mm diameter. Towards reducing errors, particularly the variation in surface friction’s influence on slump, the interior of the mould as well as its base were moistened at the start of the test. The slump measurement was set to the nearest 5 mm. The pressure method ASTM C231/M231-14 was applied to obtain the air content of fresh POFA-rubberized concrete. Type A air meter was used for the evaluation of the air content of fresh concrete. The test was started within 15 min after concrete mixing. The bowl was filled with three equal layers, and each layer was rodded 25 times. Then, the base was struck with a mallet 12 to 15 times to remove entrapped air. The flange of the measuring bowl and that of the cover were thoroughly cleaned and then clamped. Water was added over the fresh concrete until the water level reached the zero mark on the tube. The vent was closed, and the pressure was provided by a hand pump. Pressure was applied until the pressure gauge reached a calibrated point (1.4 kPa). When the pressure reached the required value, the air content of concrete was computed.

The mechanical properties of modified rubberized concrete were conducted by measuring the compressive, indirect tensile, and flexural strength according to ASTM C39/C39M-12, ASTM C496/C496M-11, and BS 1881: Part 118, respectively. The specimens were duly prepared and the surface area of the concrete was elevated by polishing using a lathe machine. Both the top and bottom sides were elevated with a maximum of 0.05 mm tolerance in elevation. The loading machine was cleaned, and the digital unit of the machine was programmed with the required sample information. A constant loading was selected and applied until the failure of the specimen.

Additionally, microstructure studies were conducted by using scanning electron microscope (SEM) and X-ray Diffraction analysis (XRD) to provide supplementary data on the physical and chemical properties of POFA and the effect on proposed concrete. SEM test was used to study the microstructure of concrete. A JOEL 7600F SEM versatile ultra-high-resolution device with variable pressure solution, equipped with an energy dispersive X-ray analyzer, was used. The magnification of JOEL 7600F was in range of 25 to 1,000,000. The 7600F was equipped with an SDD type X-ray detector system. This type of detector is a significant advance that can acquire and process more than 100,000 X-ray counts per second. To achieve clear resolution of surface structure, specimens were coated with platinum in an “Auto Fine Coater”. Drying was necessary, as specimens containing a large amount of moisture or oil will often contaminate the inside of the electron optical column. This is normally caused by deposits that evaporate from the vacuum chamber in which the test specimen is positioned. XRD analysis was conducted using a Bruker—D8 Pioneer diffractometer. The X-ray Source was a 2.2 kW X-ray tube. The running conditions for the X-ray tube were 40 kV and 40 mA. The identification of crystalline phase was computer-based using the Xpert Score software. The database is useful for identifying approximately 200,000 metal crystal phases, alloys, oxides, salts, and other compounds.

### 2.4. Experimental Design Using RSM Model

The main idea behind deploying response surface methodology is to (1) assess the relationship and interaction between the variables, and (2) predict and obtain optimum values of the influential factors that achieve the best performance of the target output [70]. Therefore, according to the central composite design (CCD), which is a type of response surface methodology, only part of the entire experimental data is used to achieve this goal, as shown in Table 5. Indeed, CCD composed of three types of points including centre points, 2*n* axial points, and 2*^n^* factorial points where *n* is the number of independent variables. Figure 4 shows the required experimental tests on the basis of that the two independent variables were used. It is interesting to note that the coded values +1 and −1 refer to the high and low limit of each parametrise, while α is the distance from centre of the square which is equal to 1.0. The number of experimental tests (*Q*) was found to be nine using Equation (1), where *m* represents the number of centre points [71]. Moreover, Equation (2) was used to obtain the coded values, where *Z* and *Zo* are the real value of independent value and real value of independent variables at the centre point, respectively [72]. Furthermore, *L* denoted the coded value of the independent variable. Table 5 refers to the CCD-required data to run all data sets.
(1)$$Q=2^{n}+2n+m$$
(2)$$L=\frac{Z-Z_{c}}{\alpha }$$

Second order polynomial equation was considered to obtain the mechanical properties of concrete. The general form of second order polynomial equation is presented in Equation (3) [73]. Where *β*_ii_ is the quadratic coefficients, *β*_o_ corresponds to intercept of the model, and *β*_i_ denotes to the linear coefficients. It should be noted that three models were developed to optimize the mechanical properties of POFA concrete incorporating rubber. In the first model, *X*_1_ and *X*_2_ represent the input data involving Fibre rubber and time, while *Y* is the response (mechanical properties). In the second model, the input parameters were fine rubber and time, while coarse rubber and time were represented in the third model. For the purpose of verification of the proposed equation, analysis of variance (ANOVA) was taken into account. In particular, *R*^2^ was calculated to evaluate the closeness between the response and real results as shown in Equation (4) [61]. Where *SS_T_* represents the total sum of square error, *SS_E_* is the sum of square error based on the predicted results, and denote the mean value of actual value. In addition, *Y_P_* and *Y_A_* are the predicted and real values. In the same context, $R_{adj.}^{2}$ also calculated to evaluate the effect of the number of independent variables on the correlation between the real and predicted results as shown in Equation (5) [74]. Where, *DF* is the degree of freedom and *SS_R_* represents the sum square of differences error between the actual and predicted values. The predicted $R_{pred.}^{2}$ was also determined according to Equation (6) [75]. The differences between $R_{pred.}^{2}$ and $R_{adj.}^{2}$ should be less than 0.2 to ensure that the equation has the ability to predict for another data set [76]. *W* refers to the estimated residual sum of square without the *i_th_*. Meanwhile, the signal-to-noise ratio was evaluated using the adequate precision (S/N) as shown in Equation (7) in which the value should be greater than 4 [77], where σ^2^ denotes to the residual mean square. In addition, *p*-value and *F*-value were also taken into account to validate the significance of the proposed equation. The achievement of high F-value and *p*-value less than 0.005 led to considering the equation is significant [78].
(3)$$Y=\beta_{o}+\sum \beta_{i}X_{i}+\sum \beta_{ii}X_{i}^{2}+\sum \beta_{ij}X_{i}X_{j}$$
(4)$$\begin{array}{ll} R^{2} & =\frac{SS_{E}}{SS_{T}}\\ & =\frac{{\sum_{i=1}^{n}\left(Y_{P}-{\bar{Y}}_{A}\right)}^{2}}{{\sum_{i=1}^{n}\left({\bar{Y}}_{A}-Y_{A}\right)}^{2}} \end{array}$$
(5)$$R_{adj}^{2}=1-\frac{SS_{R}/DF_{R}}{SS_{T}/DF_{T}}$$
(6)$$R_{pred}^{2}=1-\frac{W}{SS_{T}}$$
(7)$$S/N=\frac{\max(Y_{p})-\min(Y_{p})}{\sqrt{\frac{p\sigma^{2}}{n}}}$$

## 3. Results and Discussion

### 3.1. Workability

Slump test was conducted to evaluate the effect of rubber aggregates on the workability of POFA-modified concrete in terms of the level of rubber aggregate substitution and the types of rubber aggregates. The slump values of rubberized concrete in various amounts and types are presented in Figure 5. Incorporating fine fibre rubber aggregates into concrete resulted in a change in the slump value, and this effect was more pronounced when the percentage of rubber replacement was increased. The slump value of concrete incorporating 20% POFA (CP20) was 85 mm, and that of plain concrete (CP) was 100 mm. Fibre rubber aggregates increased the workability of concrete to 90 mm and 100 mm by incorporating 5% and 10% replacement levels, respectively. The slump values decreased when the amount of rubber replacement exceeded 10% of fine natural aggregates. The results of the slump test for concrete containing 20% and 30% fine rubber aggregates were 85 mm and 80 mm, respectively. Incorporating fine granular rubber aggregates into concrete led to the enhancement of slump values, similar to type R_1_ rubberized concrete. The graph of the slump value exhibited a bell curve, with R_2_10CP20 appearing at the peak of the graph with a slump value of 105 mm. The obtained results indicated that incorporating fine granular rubber aggregates led to the enhancement of workability when the concentration of R_2_ TRA was lower than 20% of the total natural fine aggregates. The workability of rubberized concrete was decreased when the level of rubber aggregate replacement reached 30%. For instance, the slump value of R_2_20CP20 and R_2_30CP20 was 95 mm and 85 mm, respectively. The slump of concrete containing coarse granular rubber aggregates with 1% SP were obtained as 55, 45, 30 m, and 20 mm for 5%, 10%, 20%, and 30% rubberized concrete, respectively. The slump values exhibited a greater reduction in concrete incorporating coarse rubber aggregates compared with the two other types of fine rubber aggregates. Furthermore, lower workability was observed with higher coarse rubber concentration. Finally, the lower workability of R_3_ rubberized concrete can be attributed to the roughness of rubber particles and the higher specific surface area than those of natural aggregates. This reduction may also be related to the poor cohesion between rubber aggregates, cement paste, and natural aggregates [79,80].

### 3.2. Air Content

Figure 6 shows the trend of air content variation of rubberized concrete relative to the amount and type of rubber particles. The air content results showed that the incorporation of rubber into concrete resulted in an increase in the air content of the concrete mixture. Furthermore, the geometry and size of rubber particles are important parameters affecting the air content of rubberized concrete. For instance, rubberized concrete with fine fibre rubber aggregates (R_1_) showed higher values of air content than concrete incorporating fine granular particles (R_2_). The shape of the fibre rubber aggregates plays an important role in this behavior, and the number of particles per unit volume of the fine fibre was higher than that of R_2_. R_1_ also entrapped more air during the mixing process of concrete because of the shape of fibre, such that it might easily bend during the mixing process and could thus entrap air more than the granular aggregates. Meanwhile, the air content of concrete containing coarse rubber aggregates (R_3_) showed lower values than concrete with fine rubber particles (R_1_ and R_2_). Rubber particles are known to be non-polar by nature (water insoluble). Thus, during mixing, they were able to trap air bubbles at the particle surfaces. Moreover, the specific gravity of coarse granular rubber aggregates (1.364) was higher than that of both types of fine (1.332) and fibre rubber aggregates (0.884), and the number of rubber particles in a unit was less than that for both fine rubber aggregates. Therefore, the amount of air bubbles was reduced, perhaps causing the lower air content in the R3 batches of the concrete mix [69,81].

### 3.3. Compressive Strength

Figure 7 presents the experimental and predicted results of the compressive strength of concrete containing fibre rubber aggregates (R_1_) at different curing times. When R_1_ concentration increased, lower strength development was observed as shown in Figure 7a. This is also in with the predicted results as shown in Figure 7b in which a high slop gradient was observed with the increase in rubber content for all output. This fact was also confirmed by ANOVA in which the *p*-value of fibre rubber was 0.0004, whereas the F-value of CA was greater than 255 for all data sets indicating that the effect of fibre rubber on compressive strength was significant.

It should also be noted that the compressive strength values of normal concrete were 34.43, 40.41, 45.08, 48.65, 49.78, and 51.18 MPa at 7, 28, 56, 90, 180, and 360 curing days, respectively, while the compressive strength values for 20% POFA concrete were 33.5, 41.25, 47.57, 51.2, and 54.53 MPa at the aforementioned curing periods. The gain in compressive strength of POFA concrete was higher than that of OPC, and this can be attributed to the pozzolanic reaction of POFA, which continued beyond the age of 28 days, as well as the lack of hydrated POFA particles acting as fine fillers in the concrete. A higher compressive strength reflects the high fineness of the POFA, which promotes the pozzolanic behaviour that is a major factor for strength development in POFA concrete [63,82,83].

On the other hand, the compressive strengths of concrete containing fibre rubber aggregates (R_1_) were lower than that of normal and 20% POFA concrete at all ages. An increase in rubber particle content was found to deteriorate compressive strength further. The 28-day compressive strength of R_1_ rubberized concrete was reduced by 5.3%, 15.9%, 29.8%, and 41.5% compared to the case where the level of rubber replacement was 5%, 10%, 20%, and 30% of fine natural aggregates, respectively. An interesting point is that the rate of strength development in rubberized POFA concrete further increased at 28 days. For instance, R_1_5CP20 showed much closer values of strength to that of normal concrete after 28 days. The compressive strength of rubberized concrete was found to improve with the introduction of 20% POFA to the concrete. Moreover, the values of the reduction in strength were limited to 2.7%, 13.6%, 27.8%, and 39.9% for 5% to 30% rubber replacement when compared to normal concrete.

Apart from fibre rubber to fine rubber, the experimental and theoretical results of the compressive strength of concrete containing fine granular rubber aggregates (R_2_) are presented in Figure 8. The results showed a reduction in compressive strength with increased concentration of R_2_ rubber aggregates. The initial 28-day compressive strength for normal concrete was 40.41 MPa, but the value decreased to 37.6, 30.6, 26.4, and 23.6 MPa when 5%, 10%, 20%, and 30% R_2_ tire rubber aggregates were introduced to the concrete mixture. The results showed that the strength behaviour of rubberized concrete highly depended on the substitution ratio of rubber aggregates. Furthermore, the compressive strength was found to decrease drastically with increasing tire rubber particle content. For instance, the compressive strength was reduced by 6.9%, 24.2%, 34.6%, and 41.5% when fine granular TRA were replaced at 5%, 10%, 20%, and 30% sand in concrete. The strength behaviour of concrete incorporating R_2_ rubber particles was in agreement with the common understanding that the replacement of natural aggregates with tire rubber particles results in a reduction in compressive strength. However, accurate design, proper curing of concrete, and pozzolanic material addition, such as POFA, may improve the compressive strength of rubberized concrete. Emiroglu et al. [33] reported that the compressive strength of concrete containing 0 mm to 4 mm rubber aggregates was reduced by 8.7%, 26%, and 46% with 5%, 10%, and 20% levels of replacement. Moreover, Bignossi and Sandrolin [84] found a 26% and 39% reduction in the compressive strength of self-consolidating concrete containing 20% and 30% fine rubber aggregates, respectively.

In addition, both experimental and estimated results of the compressive strength of concrete incorporating granular coarse rubber aggregates (R_3_) are illustrated in Figure 9. As expected, the results exhibited a lower compressive strength than normal concrete for all rates of substitution. The results showed a reduction in strength of approximately 19.4%, 30.6%, 36.5%, and 49.2% after 28 days curing for 5%, 10%, 20%, and 30% R_3_ replacement, respectively. The strength gain behaviour of concrete incorporating POFA and R_3_ aggregates is in agreement with the pattern of POFA concrete. However, the rate of gain strength in rubberized POFA concrete was lower than that in POFA concrete alone, but it was observed that the reduction percentages in strength of concrete incorporating 5% R_3_ aggregates were 19.4%, 19.03%, 18.84%, 18.6%, and 18.4% for 28, 56, 90, 180, and 360 days, respectively, in comparison with normal concrete at the corresponding ages. The trend of lower strength of concrete containing R_3_ rubber aggregates demonstrated that the size of rubber aggregates significantly influenced the strength of rubberized concrete. A comprehensive analysis and discussion regarding the effect of rubber aggregate size are given in the next section.

It can be inferred that, the compressive strength of rubberized POFA concrete was significantly influenced by the size, type, and proportion of rubber aggregates in concrete. The results indicated that fine fibre rubber aggregates (R_1_) showed better performance than fine granular rubber aggregates (R_2_) in terms of compressive strength. This better performance significantly continued for 10% rubber aggregate replacement, but the further addition of rubber aggregates beyond 20% resulted in the disappearance of the ascendancy of fine rubber aggregates (R_1_). For instance, the compressive strength values of R_1_30CP20 were 16.32, 24.28, 28.02, 29.2, 30.53, and 31.6 MPa for 7, 28, 56, 90, 180, and 360 days, respectively. Likewise, the results of R_2_30CP20 were 16.5, 23.6, 27.6, 28.8, 30.23, and 31.4 MPa for the corresponding days. These results indicated that the compressive strength of 30% R_1_ and R_2_ rubberized concrete showed similar performance, with the difference in compressive strength being insignificant for all ages of concrete.

This shows that the type (fibre and granular) of rubber aggregate was only important for low levels of replacement, and for medium and high amounts of proportion, the influence of the type of rubber particles on compressive strength was minimal. In other words, the deterioration of compressive strength with 30% replacement of fine natural aggregates by tire rubber particles was more significant in terms of the amount of rubber aggregate in the mixture, fibres, or granular rubber particles. Indeed, the compressive strength behaviour of concrete attributed to the incorporation of fine rubber aggregates was affected and controlled by the type of rubber aggregate for 5% and 10% replacement, but the effect of rubber type on the strength behaviour was reduced for 20% replacement. For 30% replacement, the behaviour was significantly controlled by the rubber aggregate content rather than the type of rubber particles.

The maximum amount of coarse replacement aggregate was limited to 30%, and the compressive strength could be marginally acceptable for concrete used for structural application. The strength behaviour of rubberized concrete may be explained by several possible reasons for the reduction in strength with the inclusion of coarse rubber aggregates in the concrete mixture. The TRAs were significantly softer than the surrounding cement paste and natural aggregates. Therefore, rubber aggregates showed a higher elasticity and deformability than other concrete components upon the application of load. In this case, cracks begin forming around the rubber particles. This hastened the failure of rubberized concrete. Another reason may be the non-polar nature of rubber particles, which may significantly reduce the adhesion between rubber aggregates and cement paste, thereupon creating a weak phase in the concrete. The soft rubber aggregates may be considered as voids in the concrete matrix. Furthermore, a higher degree of air contact in concrete may boost the weakness of rubberized concrete when subjected to applied loads. Moreover, some researchers, such as Eldin and Senuci [32], likened rubber aggregates to air content and voids. Thus, the lower strength of concrete incorporating coarse rubber particles can be attributed to the larger voids that effectively reduced concrete strength. In addition, the strength of concrete is strongly dependent on coarse aggregate size grading and hardness [85]. Consequently, the partial replacement of aggregates with rubber is anticipated to cause a reduction in strength.

### 3.4. Splitting Tensile Strength

The results of the splitting tensile strength of rubberized concrete are presented in Figure 10. The respective strength values were determined for 7, 28, 56, 90, 180, and 360 days for the long-term investigation of the effect of rubber aggregates on the splitting tensile strength behaviour of rubberized POFA concrete. The splitting tensile strength value of normal concrete was considered as a benchmark, and the variances in the tensile strength of rubberized concrete are presented as percentages of normal concrete. Notably, introducing POFA to the mixture resulted in a reduction in splitting tensile strength by approximately 2.25% at 28 days. Meanwhile, the gain in tensile strength beyond 28 days was greater than that of normal concrete, and the tensile strength of 20% POFA concrete was 0.8% greater than that of conventional concrete. Similar to compressive strength, the addition of rubber aggregates decreased the splitting tensile strength of rubberized concrete. A 5% replacement of fine fibre rubber aggregates (R_1_) achieved the best result among the three types of rubber aggregates. The reduction in splitting tensile strength was more pronounced with the use of coarse rubber aggregates than with fine rubber particles. The percentage of reduction was observed for all levels of replacement. The results indicated that in relation to the size and type of rubber aggregates, the fine fibre rubber particles performed better than granules in both coarse and fine particles. This was in line with the finding of Li et al. [86], who recommended the utilisation of rubber aggregates in the form of fibre rather than chips. Ganjian et al. [87] supported this conclusion as well. Similar to compressive strength, the size of rubber aggregate substantially affects tensile strength. Tensile strength was found to decline as the size of rubber particles increased. Furthermore, the splitting tensile strength of rubberized concrete decreased as the concentration of rubber particles increased in all groups of rubber aggregates. This behaviour is congruous with the compression strength observation. The main reason for the reduction in tensile strength relating to concrete incorporating rubber can be attributed to several causes. The weak bonding between cement pastes and rubber aggregate results in concrete failure starting in the weak interface zone. Another variable has to do with the nature and proportion of rubber aggregates as compared with conventional aggregates. The stiffness of rubber particles was very low, such that they were easily deformed. Therefore, the tire rubber particles are functioning just as voids and cavities, which inevitably caused stress concentration to be on the periphery of the rubber aggregate and resulted in a decline in the spitting tensile strength of modified concrete. Another important observation is that the tensile strength of POFA-rubberized concrete was more significantly influenced by the size and the content of rubber particles than the type and shape of rubber aggregates. For instance, the tensile strength of rubberized concrete in groups of R_1_ and R_2_ with 30% tire rubber particles were extremely close, such that no significant difference in tensile behaviour was found in both groups. The behaviour and trend of tensile strength in concrete with high-volume rubber content can be concluded to be independent of the type or shape of rubber aggregates in the same grade.

A visual inspection of the crushed sample showed different behaviours of rubberized concrete because the rubber particles were never cut or broken under loads; the rubber aggregates were observed in intact conditions (Figure 11). Despite the fact that normal concrete often exhibits brittle failure, rubberized concrete specimens showed ductile failure. This may be attributed to the deformability and plastic behaviour of tire rubber particles. Furthermore, the specimens of rubberized concrete were not fully opened after failure, and more loads had to be applied to separate samples into two cuts after the failure occurred. This behaviour was more pronounced when the rubber concrete content was increased. Moreover, the crushed sample of fibre-modified rubberized concrete was found to have more resistance to being fully opened than rubberized concrete containing fine granular rubber aggregates. Figure 12 shows the behaviour of concrete incorporated with different contents of rubber aggregates during the tensile strength test. Valadares et al. [88] reported the marked agreement in tensile behaviour and concluded that rubberized concrete showed a higher capacity for absorbing plastic energy during the splitting test.

### 3.5. Flexural Strength

The flexural strength of concrete containing 20% POFA and 5% to 30% tire rubber aggregate is presented in Figure 12. Incorporating 20% POFA decreased the flexural strength of the resulting concrete to approximately 9.85% and 6.83% at 28 and 56 days, respectively. The flexural strength for CP20 reached 6.4, 6.53, and 6.76 MPa at 90, 180, and 360 days, respectively, thus showing a 1.71%, 0.61%, and 0.44% reduction when compared with plain concrete. The incorporation of fine fibre rubber aggregates decreased the flexural strength because of the weak bond between the cement paste and rubber aggregates.

The reduction in flexural strength was not as significant as observed in terms of the reduction in compressive strength because of the incorporation of TRA. Furthermore, the substitution of fine fibre rubber (R_1_) aggregates yielded concrete with poorer performance than that with granular particles (R_2_ and R_3_), regardless of the gradient particle size for all levels of replacement. This was exactly opposite of the trend of the compressive strength of concrete with these three types of aggregates. The performance of R_1_ and R_2_ in terms of flexural strength was very close. For instance, flexural strength was 5.35 (−14.94%) and 5.77 (−9.8%) at 28 and 90 days, respectively, for R_1_5CP20. Meanwhile, these values were 5.61 (−10.81%) and 5.96 MPa (−6.9%) for R_2_5CP20.

Fine granular TRA showed better results at 5% replacement among the three types of rubber aggregates. Meanwhile, introducing R_1_ showed lower values of flexural strength. A similar trend was observed for the 30% levels of replacement. In general, rubber particles are materials with high deformability properties and can be considered as a barrier against crack growth in concrete. Therefore, this material was expected to improve the flexural strength of rubberized concrete. However, the results showed the opposite of this hypothesis. According to the theory of flexure, when a concrete sample is under flexural stress, the concrete produces tensile stress on one side of its neutral axis and compressive stress on the other. When concrete has low tensile strength as compared to its compressive strength, failure will occur before the concrete reaches its ultimate strength in the compression area. Consequently, the major factor contributing to the decline in flexural strength, similar to that for compressive strength, was the weak bonding between the TRA and cement paste. The visual observation of crushed samples supported this conclusion, given that the rubber particles were not crushed and were easily removed from concrete. Notably, the failure mode of rubberized concrete under flexure showed greater flexibility than normal concrete. This attribute was more pronounced with the increasing amount of rubber particles in the three types of rubber concrete. As shown in Figure 13, the failure mode of concrete with no rubber aggregate was blast (brittle) failure, and the concrete was divided in two pieces. Meanwhile, rubberized concrete resisted full failure, and only a crack appeared in the middle of the beam specimens. More load is needed to be applied to the specimens to break them in two pieces.

### 3.6. RSM Model Verification

The evolution of mechanical properties of concrete incorporating fibre, fine, and coarse rubber was developed using quadratic equations as shown in Table 6. These predictive equations could also be used to quickly provide insights into the interaction and effect of rubber on the compressive, flexural, and tensile strength of POFA concrete. It can be seen that all equations are functions of time and rubber. Based on ANOVA results, the developed equations of mechanical properties of POFA concrete incorporating fibre, fine, and coarse rubber were found to be significant in which F-values were greater than 272.64, 78.65, and 158.23, respectively, while *p*-values were less than 0.0003, 0.0022, and 0.0008, respectively. This is consistent with the previous studies who demented that the model could be considered as significant if the *p*-value is lesser than 0.005 and F-value is high [89]. In the same contest, *R*^2^ proved the closeness and correlation between the predicted and actual results in which the value of *R*^2^ was high. According to Huseien, Sam [53], a good correlation could be obtained when *R*^2^ is greater than 0.7. Herein, the *R*^2^ value of mechanical properties of POFA concrete incorporating fibre, fine, and coarse rubber were greater than 0.9978, 0.9924, and 0.9962 highlighting that the predicted results were acceptable. In addition, the capability of these quadratic equations to accurately predict further data was also proved using $R_{prdicted}^{2}$ and $R_{adj}^{2}$. In particular, it was found that the differences between $R_{prdicted}^{2}$ and $R_{adj}^{2}$ were less than 0.2. This fact is a good agreement with the existing literature. For example, Jitendra and Khed [90] developed the RSM model to predict and optimize the water absorption, chloride ions penetration, and compressive strength of concrete blocks containing foundry sand and fly ash as partial replacements of natural sand and cement, respectively. The outcome of their study revealed that the model was reliable and could be used for further predictions. This is because a reasonable difference (less than 0.2) between $R_{prdicted}^{2}$ and $R_{adj}^{2}$ was achieved for all data sets.

### 3.7. XRD Results

The X-ray diffraction patterns of raw materials of OPC and POFA are depicted in Figure 14a,b. The major mineral presenting a crystalline structure was quartz. POFA was found to have a quartz intensity of about 3500 counts. Furthermore, POFA contains opal, the formation of which resulted from the calcination of the organic constituents of palm oil fibres and shells. Similar results were reported by Altawir et al. [91] and Hassan et al. [29]. As shown in Figure 14c, the influence of POFA addition in the concrete matrix revealed pronounced diffraction peaks at 2*θ* values in the range from 16 to 30°, which were assigned to crystalline silica and alumina compounds, respectively. Nonetheless, the occurrence of other crystalline peaks was allocated to the presence of crystalline quartz and mullite phases. However, incorporation of POFA was required to overcome the low SiO_2_ content in OPC. Calcium–silicate–hydrate or C–S–H is formed from the reaction between SiO_2_ and Al_2_O_3_ in a pozzolanic material with Ca(OH)_2_ in a cement paste. Ca(OH)_2_ acts as an indicator for pozzolanic reactions. Chindaprasirt et al. [92] reported that as pozzolanic replacement and fineness increases, Ca(OH)_2_ content decreases while fortifying the sulphate resistance in the concrete. The pozzolanic reaction of high-fineness POFA is denser and more homogenous, enabling it to increase the compressive strength of concrete.

### 3.8. Scanning Electron Micrographs

Figure 15 exhibits the effect of the fine, coarse, and fibre TRAs on the bond zone between the rubber aggregates and cement paste. The change of TRAs particle size as natural aggregates replacement affected the bond zone between the rubber aggregates and cement paste and created a reduction in strength. This reduction in strength can be ascribed to the bond defects between the rubber aggregates and cement paste [93]. Corinaldesi et al. [94] reported that the drop in strength of the rubberized mortar containing stirene butadiene rubber particles (SBR) can be linked to the weak bonding between cement paste and rubber aggregate, resulting in concrete failure starting in the interfacial transition zone (ITZ). The number and size of cracks are influenced by the TRAs size. The R_1_30CP20 specimens (24.3 MPa) presented lower numbers of cracks with smaller size compared to the R_2_30CP20 (23.6 MPa) and R_3_30CP20 (20.5 MPa) specimens; with a wider bond zone, the compressive strength of concrete containing coarse TRAs was lower than that of concrete containing fine TRAs, regardless of whether the shape was fibre or granular. For the fine TRAs, it was concluded that concrete containing fine fibre rubber aggregates performed better than that incorporating fine granular rubber at lower than 30% replacement. A possible explanation for the difference in performance between granular and fibre particles may be attributed to the different load transferring capabilities between granular and fibre rubber aggregates. Once the load is applied to rubberized concrete, granular rubber particles do not have enough length to transfer the applied load through interfacial frictional force, whereas fibres have a longer length to transfer the applied load, thus resulting in higher strength.

## 4. Conclusions

This section presents the outcomes of the research.

The workability of rubberized concrete made with fine and fibre TRAs is much closer to that of conventional concrete if the rubber replacement level is limited to 10%. Increasing the size and percentage of TRAs decreased the workability due to the increased friction by rubber particles.The air content in R_1_30CP20 was 2.88 times higher than that of concrete containing 20% POFA alone, indicating that the air content increased by increasing the fraction of rubber aggregate, which is attributed to the geometry and specific gravity of rubber particles.A significant decrease in strength with a higher level of rubber replacement especially beyond 20% can be ascribed to the softer and non-polar nature of TRAs which leads to reducing the cohesion of the concrete matrix. Meanwhile, the influence of the type of rubber particles on strength was minimal.The compressive strength of POFA-influenced concrete started to exceed that of OPC concrete at 28 days by 2%, confirming that the pozzolanic reaction of POFA is less influential at early ages and increases with time.The pozzolanic reaction of POFA enables the increase in the strength of concrete, as a consequence of denser concrete by generating more (C–S–H) gel and enhancing the bond between the TRA particles and cement paste.The non-linear equations proposed here proved their ability to predict the compressive, tensile, and flexural strength with minimum error and high correlation between the actual and predicted data (*R*^2^ > 0.99, R > 0.994), thus confirming both the robustness and reliability of the models.A reasonable difference (less than 0.2) between $R_{predicted}^{2}$ and $R_{adj}^{2}$ was achieved for all data sets. The predicted mechanical properties of POFA concrete incorporating TRAs were consistent with the actual result in which a minimum error and high correlation were obtained, indicating that the models could be used for further observation in the future.

## Figures and Tables

**Figure 1 polymers-14-02294-f001:**
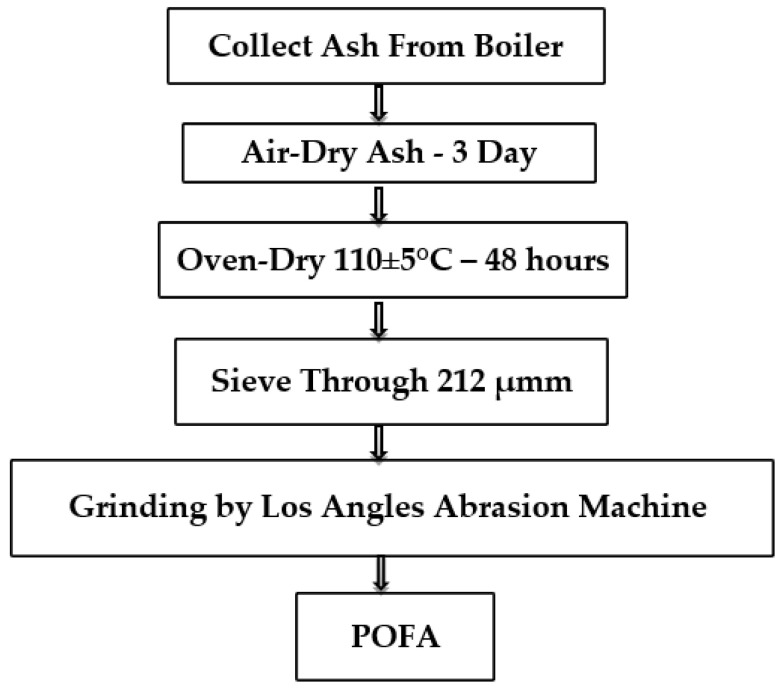
POFA preparation process for use in concrete.

**Figure 2 polymers-14-02294-f002:**
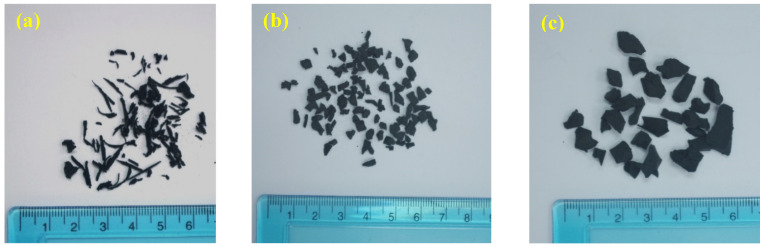
(**a**) Fine fibre rubber aggregate—Type R_1_, (**b**) Fine granular rubber aggregate—Type R_2_, (**c**) Coarse granular rubber aggregate—Type R_3_.

**Figure 3 polymers-14-02294-f003:**
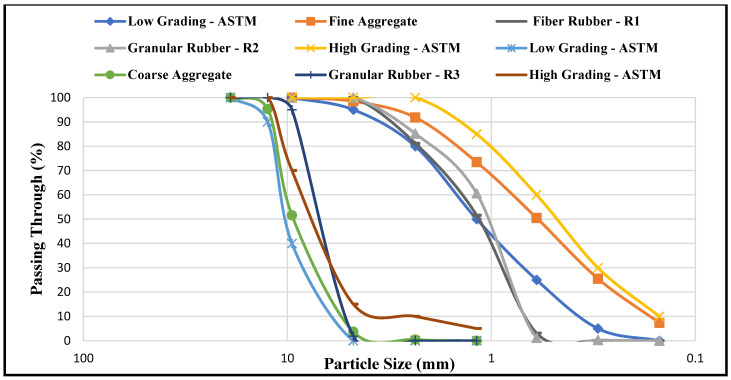
Grading Curve of natural and tire rubber aggregates in relation to ASTM C33 limits.

**Figure 4 polymers-14-02294-f004:**
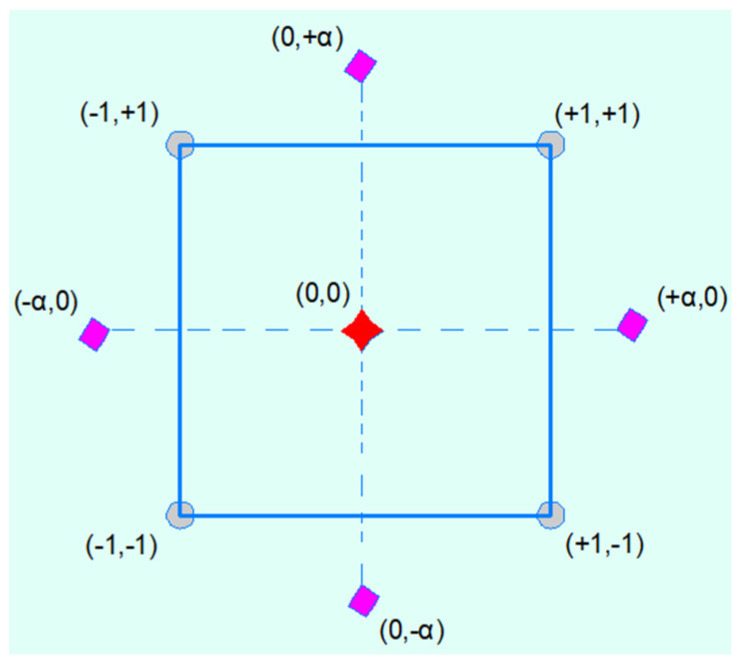
Basic concept of experimental design using CCD-RSM model.

**Figure 5 polymers-14-02294-f005:**
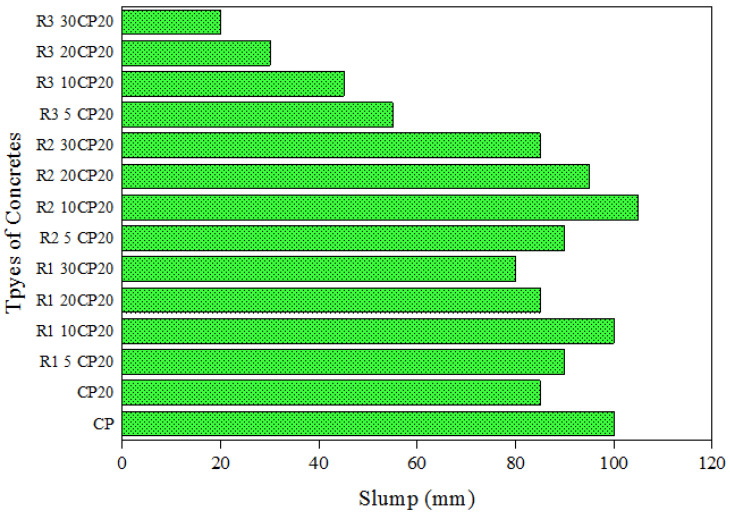
Slump values of concrete with different types and concentrations of TRAs.

**Figure 6 polymers-14-02294-f006:**
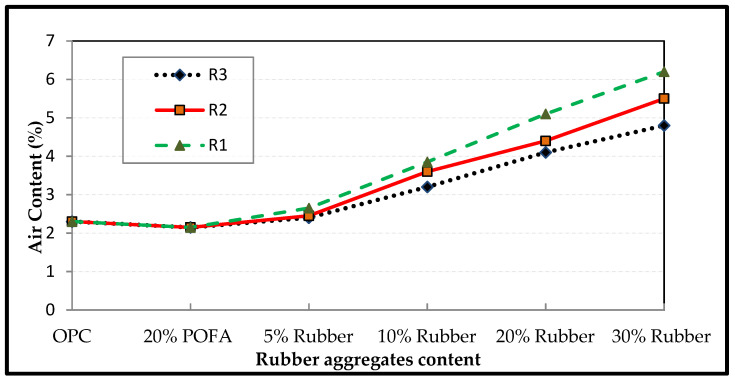
Effect of types and concentrations of TRAs on Air content.

**Figure 7 polymers-14-02294-f007:**
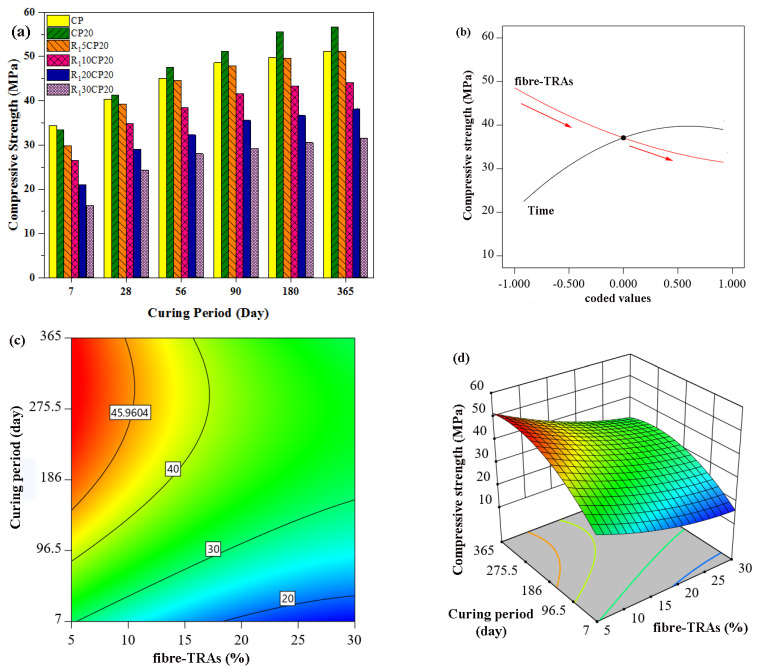
Effect of POFA concrete containing fibre-TRAs on compressive strength, (**a**) Experimental work, (**b**–**d**) RSM model.

**Figure 8 polymers-14-02294-f008:**
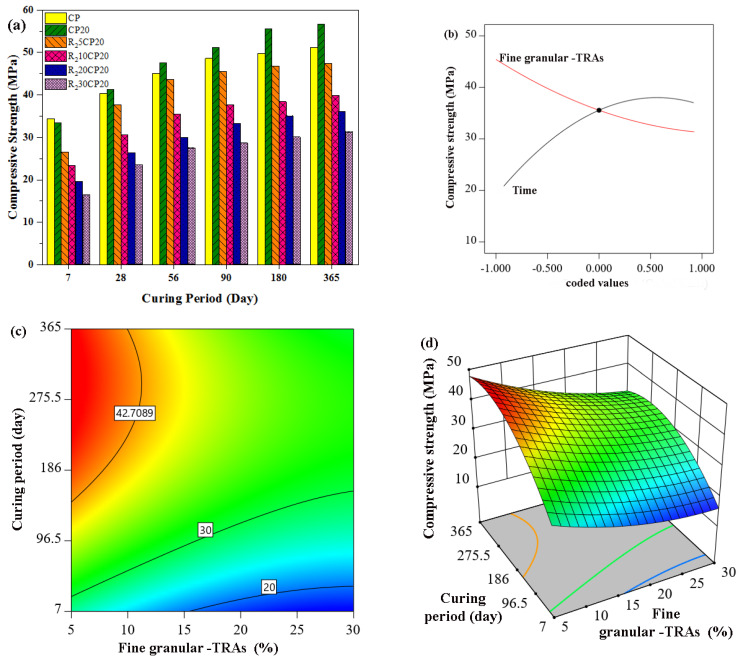
Effect of POFA concrete containing fine granular-TRAs on compressive strength, (**a**) Experimental work, (**b**–**d**) RSM model.

**Figure 9 polymers-14-02294-f009:**
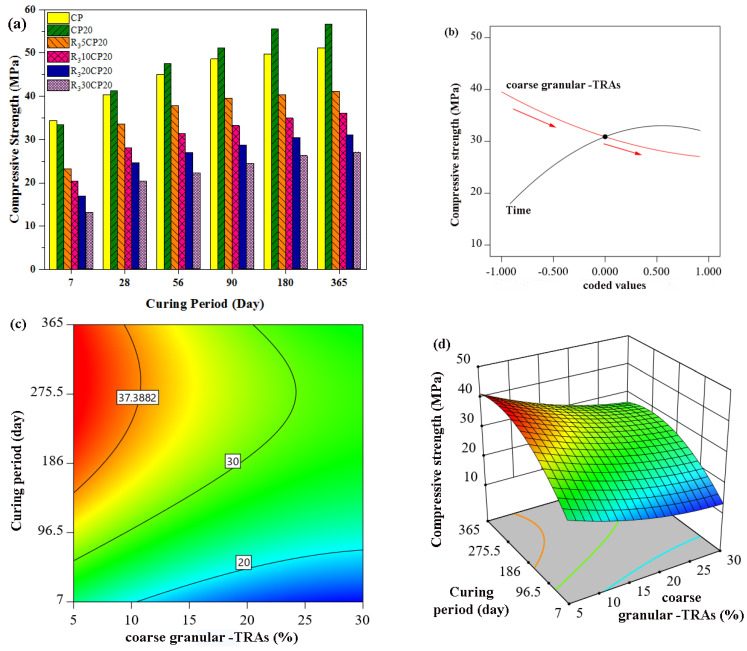
Effect of POFA concrete containing coarse granular-TRAs on compressive strength (**a**) Experimental work (**b**–**d**) RSM model.

**Figure 10 polymers-14-02294-f010:**
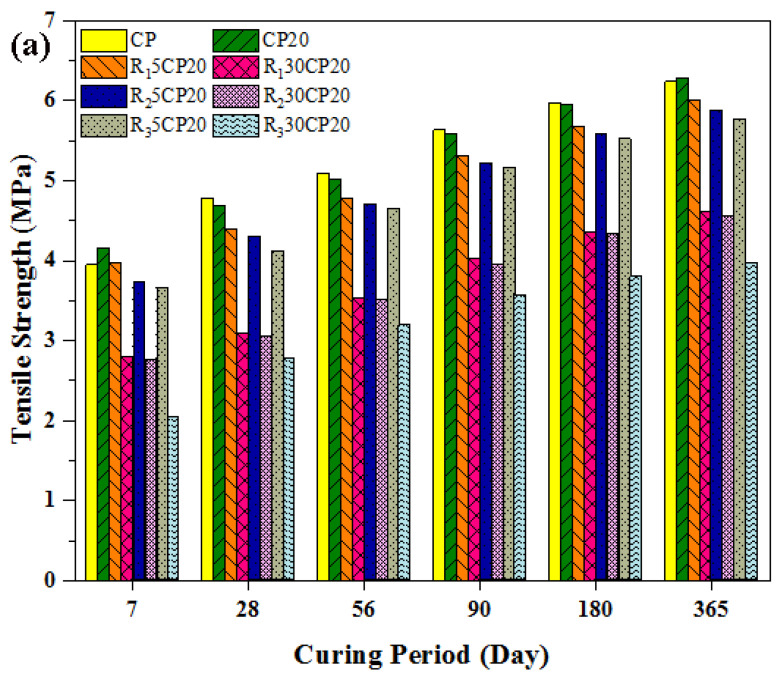
Effect of different types and concentrations of TRAs on splitting tensile strength in POFA modified concrete, (**a**) Experimental work, (**b**,**c**) fibre rubber, (**d**,**e**) fine rubber, and (**f**,**g**) coarse rubber.

**Figure 11 polymers-14-02294-f011:**
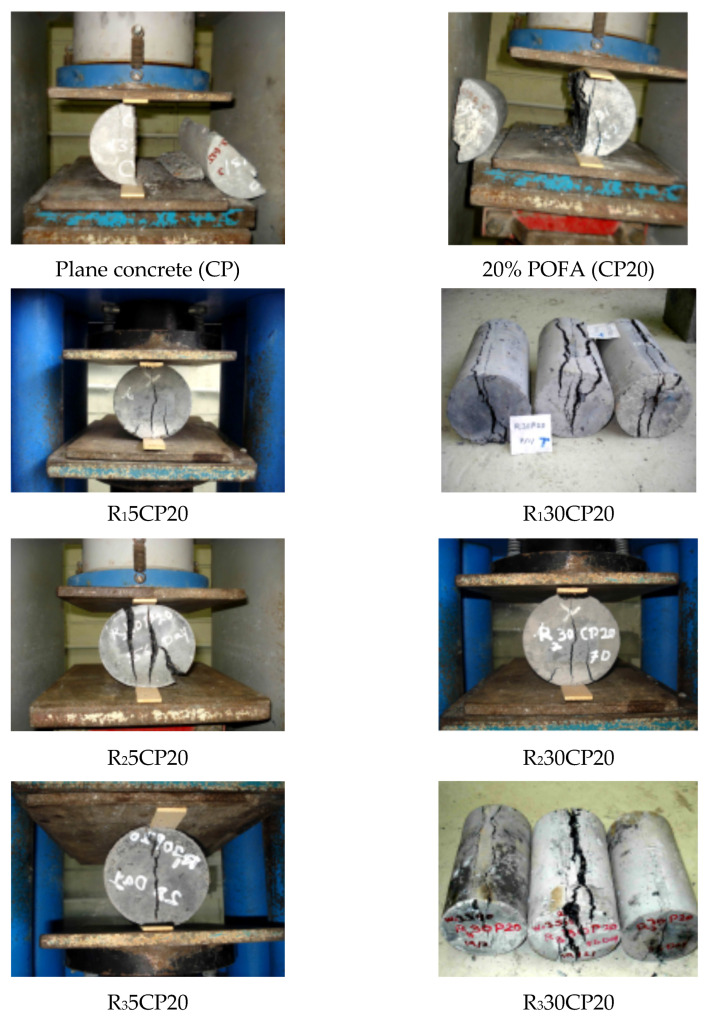
Rubberized concrete sample showing large deformation without disintegration by increasing the level of rubber particles in mixture.

**Figure 12 polymers-14-02294-f012:**
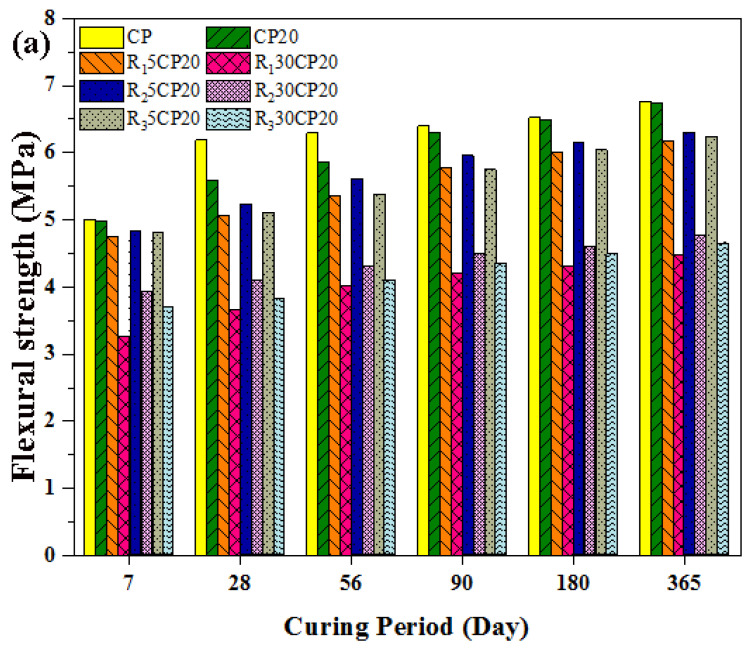
Effect of different types and concentrations of TRAs on flexural strength in POFA-modified concrete, (**a**) Experimental work, (**b**,**c**) fibre rubber, (**d**,**e**) fine rubber, and (**f**,**g**) coarse rubber.

**Figure 13 polymers-14-02294-f013:**
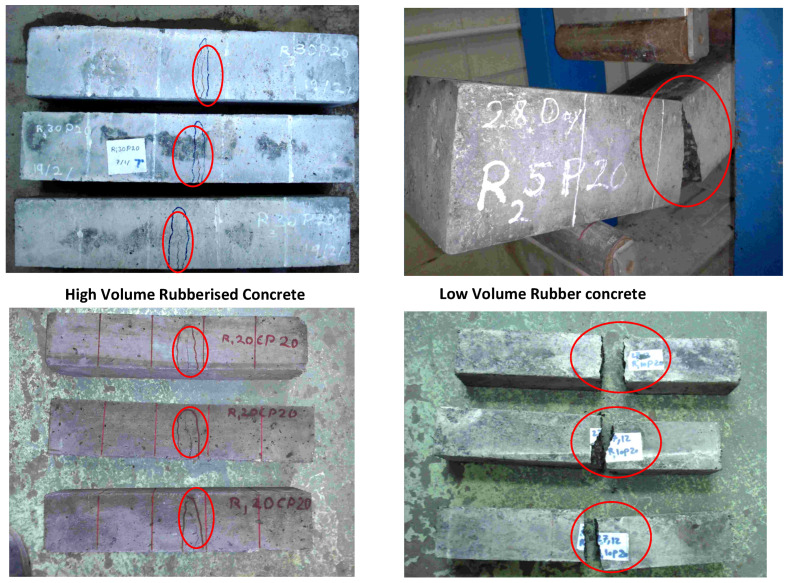
Failure mode of concrete under flexural test of rubberized POFA concrete containing high (20% and 30%) and low (5% and 10%) volumes of rubber aggregates.

**Figure 14 polymers-14-02294-f014:**
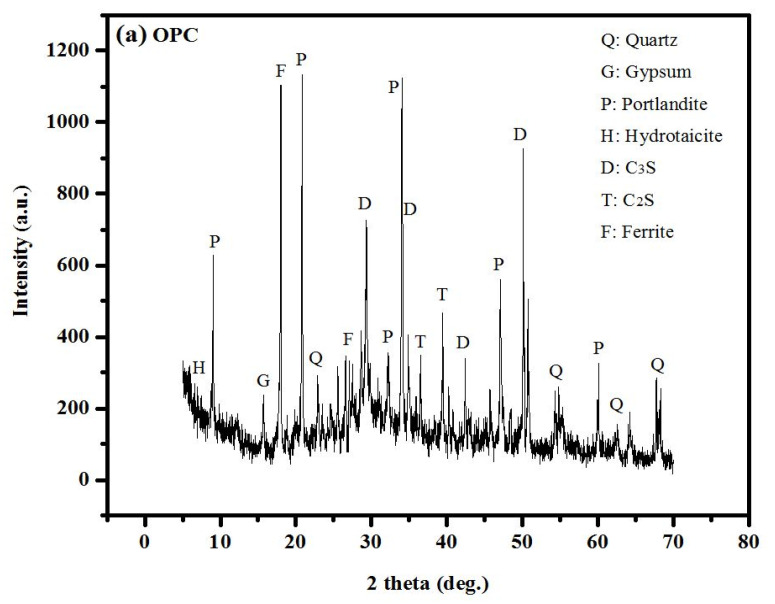
XRD pattern of (**a**) cement raw material, (**b**) POFA ash powder, and (**c**) cement pastes of binary blend.

**Figure 15 polymers-14-02294-f015:**
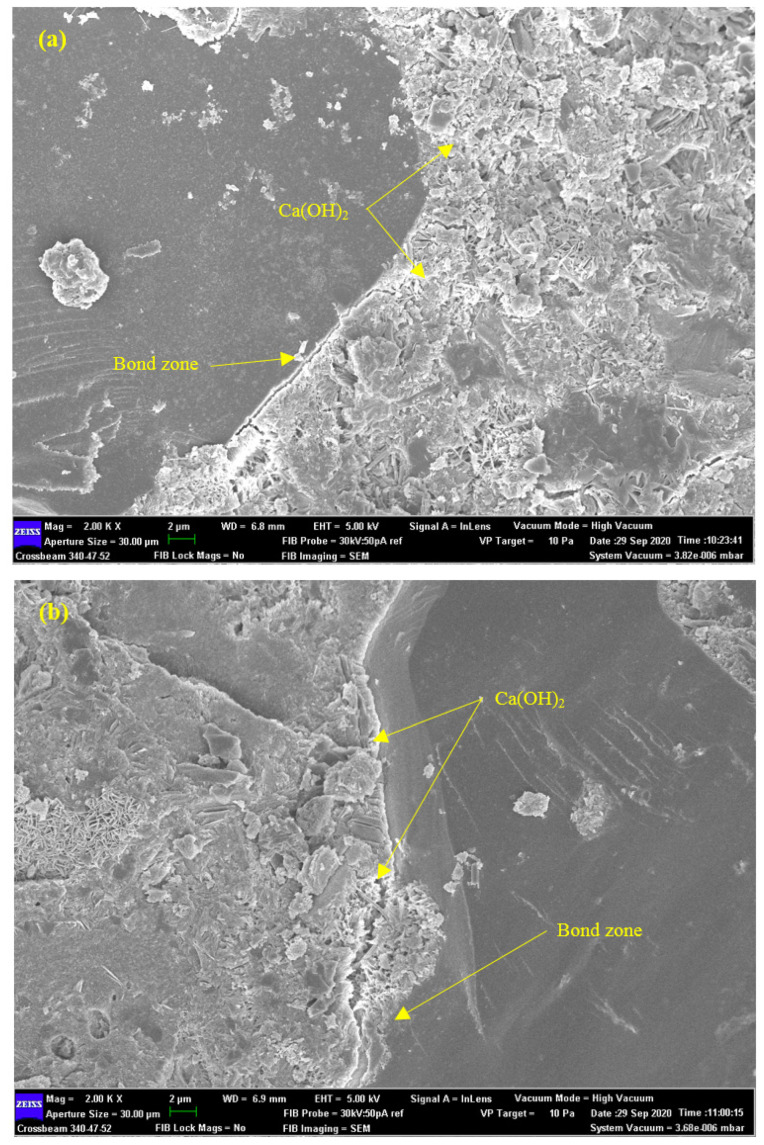
SEM image of (**a**) R_1_30CP20, (**b**) R_2_30CP20, and (**c**) R_3_30CP20 specimens.

**Table 1 polymers-14-02294-t001:** Physical and chemical properties of OPC and POFA.

Physical Properties	OPC	POFA
Specific gravity	3.15	2.43
Particle retained on 45 µm sieve	4.58%	0.73%
Median particle d10 (µm)	2.86	1.32
Median particle d30 (µm)	9.13	5.24
Median particle d60 (µm)	20.09	14.69
Specific surface (cm^2^/g)	5137.11	7796.17
Strength Activity Index (%)		
At 7 days	-	98.6
At 28 days	-	103.4
**Chemical combination**	**OPC—Mass content (%)**	**POFA—Mass content (%)**
SiO_2_	16.40	63.70
Al_2_O_3_	4.24	3.68
Fe_2_O_3_	3.53	6.27
CaO	66.85	5.97
K_2_O	0.22	9.15
MgO	2.39	4.11
CO_2_	0.10	0.10
SO_3_	4.39	1.59
Cl	-	0.50
TiO_2_	-	0.30
LOI	1.67	4.46
SiO_2_ + Al_2_O_3_ + Fe_2_O_3_	-	73.65

**Table 2 polymers-14-02294-t002:** Physical and Chemical properties of rubber particles.

Physical Properties	Unit	Values
Size (ASTM D5644)	mm	Fibre 0.84–3.36, Granules 1–4 and 5–8
Heat Loss (ASTM D1509)	% in mass	<1
Metal Content (ASTM D5603)	% in mass	<0.5
Fibre Content (ASTM D5603)	% in mass	<1
**Chemical Composition**	**Values (percent in mass)**	
Acetone Extract (ISO 1407:2009)	10 ± 3	
Ash Content (ISO 247:2006)	6 ± 1.7	
Carbon Black (ISO 1408:1995)	14 ± 8	
Rubber Hydrocarbon (RHC)	52 ± 5	

**Table 3 polymers-14-02294-t003:** Specific of used super plasticizer.

Colour	Dark-Brown Liquid
Specific Gravity	1.210 at 25 °C
Chloride Content	Chloride-free to BS 5075: Part 1 and 3
Freezing Point	0 °C—can be reconstituted if stirred after thawing
Air-entrainment	Maximum 1%

**Table 4 polymers-14-02294-t004:** Rubberized concrete mix design incorporating tire rubber aggregates with water cement ration 3.8 and superplasticizer 1%.

Mixes		Binder		Tire Rubber Aggregates			Natural Aggregates	
		OPC (kg/m^3^)	POFA (kg/m^3^)	Fine Fibre Rubber (kg/m^3^)	Fine Granular Rubber (kg/m^3^)	Coarse Rubber Granular (kg/m^3^)	River Sand (kg/m^3^)	Crushed Stone (kg/m^3^)
Batch A	CP	450	-	0	0	0	782	874.5
	CP20	405	135	0	0	0	0	0
Batch B	R_1_5 CP20	405	135	13.17	0	0	742.8	874.5
	R_1_10 CP20			26.35			703.6	
	R_1_20 CP20			52.68			625.25	
	R_1_30 CP20			79			546.9	
Batch C	R_2_5 CP20	405	135	0	19.85	0	742.8	874.5
	R_2_10 CP20				39.7		703.6	
	R_2_20 CP20				79.4		625.25	
	R_2_30 CP20				119.1		546.9	
Batch D	R_3_5 CP20	405	135	0	0	22.23	782	830.77
	R_3_10 CP20					44.46		787.1
	R_3_20 CP20					88.93		699.6
	R_3_30 CP20					133.92		612.1

R_1_5-30 (fine fibre rubber aggregates replacement), R_2_5-30 (fine granular rubber aggregates replacement), R_3_5-30 (coarse rubber granular aggregates replacement), CP control mix, CP20 cement (OPC) replaced by 20% POFA.

**Table 5 polymers-14-02294-t005:** Required experimental run according to CCD for all data sets.

Run NO.	Coded Value		Real Value (%)		CCD Division
			Rubber Content %	Age (Days)	
1	−1	−1	5	7	Factorial points (2*^n^*)
2	1	−1	30	7	
3	−1	1	5	365	
4	1	1	30	365	
5	1	0	5	180	Axial points (2*n*)
6	−1	0	30	180	
7	0	−1	20	7	
8	0	1	20	365	
9	0	0	20	180	Centre points

**Table 6 polymers-14-02294-t006:** Mathematical equations of the mechanical properties of POFA concrete containing fibre, fine, and coarse rubber.

	Type of Test	
Mechanical properties of POFA concrete incorporating fibre rubber	Compressive strength (*CS*)	$\begin{array}{l} cS=34.97-1.2X_{1}+0.15X_{2}-0.00069X_{1}X_{2}+0.0018X_{1}^{2}-0.00024X_{2}^{2} \end{array}$
	Flexural strength (*FS*)	$\begin{array}{l} FS=5.44-0.17X_{1}+0.01X_{2}-0.000025X_{1}X_{2}+0.0032X_{1}^{2}-0.000016X_{2}^{2} \end{array}$
	Tensile strength (*TS*)	$\begin{array}{l} TS=4.32-0.1X_{1}+0.014X_{2}-0.000026X_{1}X_{2}+0.0015X_{2}^{2}-0.000022X_{2}^{2} \end{array}$
Mechanical properties of POFA concrete containing fine rubber	Compressive strength (*CS*)	$\begin{array}{l} CS=31.24-1.06X_{1}+0.15X_{2}-0.00067X_{1}X_{2}+0.018X_{1}^{2}-0.00024X_{2}^{2} \end{array}$
	Flexural strength (*FS*)	$\begin{array}{l} FS=5.44-0.13X_{1}+0.009X_{2}-0.00007X_{1}X_{2}+0.0025X_{1}^{2}-0.000013X_{2}^{2} \end{array}$
	Tensile strength (*TS*)	$\begin{array}{l} TS=4.13-0.103X_{1}+0.014X_{2}-0.00004X_{1}X_{2}+0.0018X_{1}^{2}-0.00002X_{2}^{2} \end{array}$
Mechanical properties of POFA concrete containing coarse rubber	Compressive strength (*CS*)	$\begin{array}{l} CS=27.36-0.95X_{1}+0.13X_{2}-0.00045X_{1}X_{2}+0.015X_{1}^{2}-0.00022X_{2}^{2} \end{array}$
	Flexural strength (*FS*)	$\begin{array}{l} FS=5.35-0.12X_{1}+0.009X_{2}-0.000054X_{1}X_{2}+0.002X_{1}^{2}-0.000012X_{2}^{2} \end{array}$
	Tensile strength (*TS*)	$\begin{array}{l} TS=3.89-0.08X_{1}+0.016X_{2}-0.00002X_{1}X_{2}-0.00044X_{1}^{2}-0.000026X_{2}^{2} \end{array}$

## Data Availability

Data sharing not applicable.

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
