# Peer review of "Systematic Experimental Assessment of POFA Concrete Incorporating Waste Tire Rubber Aggregate"

_polymers, 2022, doi:10.3390/polym14112294_

Round 1

Reviewer 1 Report

Reviewers' comments:

Manuscript ID: polymers-1734786

Full Title: Systematic Experimental Assessment of POFA Concrete Incorporating Waste Tire Rubber Aggregate.

The manuscript describes the Systematic Experimental Assessment of POFA Concrete Incorporating Waste Tire Rubber Aggregate. The manuscript needs a detailed editing. Some markings are made to just illustrate the extent of editing needed.

The authors need to consider the following comments

- Add more suitable keywords.

- The introduction section should be improved; more related papers must be discussed and critical improvement in this study must be clarified.

- Line number 68 – author should check - ……… enhance the acoustic and ductility properties in concrete. [23, 24], and also bring…..

- Figure 3 – not clear make clear.

- 2.3. Experimental program- section should be detailed.

- Please provides the references for all equations and formula.

- Figure 14: The XRD pattern of the (a) 100% OPC (b) 100% POFA and (c) 0% and 20% POFA of prepared concretes. Not clear make clear.

- SEM: how the energy of the accelerator beam used?

- 3.8 Scanning Electron Micrographs - section should be detailed.

- Conclusions: the authors need to improve with more specific results and conclusions.

- Several faults: are added or missing spaces between words: see manuscript file.

- And make all references in same format for volume number, page number and journal name.

So that I recommended this manuscript to major revision and for future process.

Author Response

The authors would like to thank the respected reviewer for the insights and useful feedback. Your suggestions and critical comments are highly appreciated. We have carefully addressed each comment as explained in the attached response to comments. All changes in the revised manuscript have been made  in red font color and can be easily retrieved.

Reviewer 2 Report

The manuscript entitled “Systematic Experimental Assessment of POFA Concrete Incor-2 porating Waste Tire Rubber Aggregate” introduced interesting results. However, some comments below should be addressed:

  • In the abstract, the goal of the paper sentences in lines 28-31 is so long.
  • In the introduction, the last paragraph is also so long, and the contribution of this work is not clear in this paragraph.
  • Sentences in lines 99-103 should be in the methodology.
  • Figure 3 is presented in your previous work. please add the citation.
  • Please add a diagram of how to link RSM to perform the experiments.
  • In the RSM model figures in the counter figures (figures 8(c), 9(c), 10(b)) the y axis should be "curing period (day)".
  • Figure 14 needs to improve in resolution.

Author Response

(The authors gave the same response as above.)

Round 2

Reviewer 1 Report

The authors have improved the revised manuscript significantly, I recommend acceptance.